# MuGSI: Distilling GNNs with Multi-Granularity Structural Information for Graph Classification

## ABSTRACT

Recent works have introduced GNN-to-MLP knowledge distillation (KD) frameworks to combine both GNN's superior performance and MLP's fast inference speed. However, existing KD frameworks are primarily designed for node classification within single graphs, leaving their applicability to graph classification largely unexplored. Two main challenges arise when extending KD for node classification to graph classification: (1) The inherent sparsity of learning signals due to soft labels being generated at the graph level; (2) The limited expressiveness of student MLPs, especially in datasets with limited input feature spaces. To overcome these challenges, we introduce MuGSI, a novel KD framework that employs Multi-granularity Structural Information for graph classification. Specifically, we propose multi-granularity distillation loss in MuGSI to tackle the first challenge. This loss function is composed of three distinct components: graph-level distillation, subgraph-level distillation, and node-level distillation. Each component targets a specific granularity of the graph structure, ensuring a comprehensive transfer of structural knowledge from the teacher model to the student model. To tackle the second challenge, MuGSI proposes to incorporate a node feature augmentation component, thereby enhancing the expressiveness of the student MLPs and making them more capable learners. We perform extensive experiments across a variety of datasets and different teacher/student model architectures. The experiment results demonstrate the effectiveness, efficiency, and robustness of MuGSI. Codes are publicly available at: **https://github.com/uyfhtdrwww2024/MuGSI.**

## CCS Concepts

• **Information systems** → *Data mining*; • **Computing methodologies** → **Supervised learning**; **Neural networks**.

## Keywords

Graph neural networks, Knowledge distillation, Graph classification

**ACM Reference Format:**
Anonymous Author(s). 2023. MuGSI: Distilling GNNs with Multi-Granularity Structural Information for Graph Classification. In *Proceedings of ACM Conference (Conference'17)*. ACM, New York, NY, USA, 12 pages. https://doi.org/XXXXXXX.XXXXXXX

|  | PROTEINS | BZR | DD | IMDB-B |
|---|---|---|---|---|
| GIN | 79.25±3.22 | 93.09±1.89 | 77.67±2.86 | 79.60±3.02 |
| MLP | 72.61±2.98 | 79.26±1.50 | 73.59±2.90 | 77.11±2.76 |
| $GLNN_{MLP}$ | 72.96±2.54 | 79.51±1.94 | 74.49±2.94 | 77.58±3.27 |

**Table 1: Experiment results for soft logits-based KD method. Here the student is MLP and the teacher is GIN[52]. Details about experiment setting can be found in Section 5.1.**

## 1 INTRODUCTION

In recent years, Graph Neural Networks (GNNs) have emerged as a powerful tool for graph-structured data and have consistently achieved superior performance in graph-related tasks in a variety of domains, such as bioinformatics [16], social network analysis [12] and personalized recommendation [21]. Building on this, GNNs are highly relevant to the *Graph Algorithms and Modelling for the Web*.

To facilitate the deployment of latency-sensitive applications, several works [45, 49, 51, 61] employ Knowledge Distillation (KD) [22] to transfer the learned knowledge from a well-trained teacher GNN model to a student MLP model, combining GNN's superior performance with MLP's fast inference speed. However, existing GNN-to-MLP KD methods mainly focus on node classification, and its application to graph classification is largely overlooked. This gap is significant as KD for graph classification presents unique challenges that are fundamentally distinct from those in node classification: (1) *Sparse learning signals.* For node classification, dense learning signals can be generated through node-level gradient updates using soft labels, especially for large-scale graphs that consist of thousands or even millions of nodes. Conversely, graph classification inherently provides sparse learning signals, as soft labels are obtained at the level of entire graphs, making the KD process for graph classification more challenging; (2) *Limited expressive power of MLPs.* Previous work [7, 61] has established that a key factor for the success of KD for node classification is the small gap in the number of equivalence classes generated by GNNs and MLPs due to the enormous input feature space of the real-world node classification datasets (more details can be found in the Appendix D of [61]). However, this condition is often not met in graph classification tasks due to the limited input feature space, which severely limits the expressive power and learning capability of student MLPs. The empirical results illustrated in Table 1 also align with our analysis, i.e., due to the outlined challenges, a GNN-to-MLP KD framework effective for node classification only yields slight gains for graph classification. Here, we adopt $GLNN_{MLP}$ as the KD framework, our implementation is similar to the one from GLNN [61], except that a graph pooling function is utilized to obtain a graph-level representation.

**Present Work.** In this work, we introduce a novel Knowledge Distillation framework titled **MuGSI** (**Mu**lti-**G**ranularity **S**tructural **I**nformation for Graph distillation) to address the aforementioned

challenges, namely *sparse learning signals* and *limited expressive power of MLPs*. (1) To tackle the first challenge, we propose multi-granularity distillation loss to align multiple distributions across various scales of graph structures between the teacher model and the student model (as discussed in Appendix A.4). Our intuition is that both local and global structural information play a critical role in graph classification as GNNs first encode a rooted subtree for each node to capture the local substructures, then a graph pooling function is utilized to obtain a whole-graph representation, which captures the global structures. The proposed multi-granularity distillation loss in MuGSI is composed of three distinct components: graph-level distillation, subgraph-level distillation, and node-level distillation. Each component targets a specific granularity of the graph structure, ensuring a comprehensive transfer of structural knowledge from the teacher model to the student model. By leveraging this multi-granularity approach, we can provide dense learning signals during the KD process and facilitate the effective transfer of structural knowledge. (2) To tackle the second challenge, MuGSI proposes to incorporate a node feature augmentation component, thereby enlarging the input feature space and enhancing the expressiveness of the student MLPs to make them more capable learners. We further utilize a specific type of Graph-Augmented MLP (GA-MLP) as a more expressive student. Notably, the time complexity of the GA-MLP is almost identical to that of a traditional MLP.

Our work also reveals the multifaceted advantages of employing KD for graph classification, addressing key challenges in computational efficiency, robustness, and resource constraints: (1) Recently, there is a line of work aiming to improve the model expressiveness [3, 5, 6, 11, 30, 33–35, 37, 44, 46, 56, 60, 63], but they are usually costly in computational time and memory space. An effective KD framework can mitigate these issues by training a lightweight student model that retains, or even surpasses the performance of a more complex teacher model. (2) Graphs are often dynamically changed, leading to distribution shifts that can adversely affect model performance at test time. Our experiments validate that an effective KD framework can serve as a potent technique to address test-time distribution shifts. (3) In dynamic environments, student MLP-type models enable incremental computation, thus significantly improve the inference speed, which facilitates the inference in CPU machines and environments with limited computational resources.

Our contributions can be summarized as follows:

- We identify an under-explored problem: the GNN-to-MLP distillation for graph classification. Furthermore, we offer an analysis explaining why existing GNN-to-MLP KD frameworks are suboptimal for graph classification tasks.
- We propose MuGSI, the first GNN-to-MLP KD framework for graph classification to the best of our knowledge, which facilitates efficient structural knowledge distillation at multiple granularities.
- We perform extensive experiments across a variety of datasets, where the results validate MuGSI's effectiveness, efficiency, and robustness. Additionally, MuGSI effectively addresses test-time distribution shifts and enables efficient inference in dynamic settings, with the student GA-MLP model being 17.18x faster than the teacher GIN model.

## 2 PRELIMINARY

### 2.1 Notations and Problem Definition

We use $\{\}$ to denote sets. The index set is denoted as $[n] := \{1, \cdots, n\}$. Throughout this paper, we consider simple undirected graphs $G = (\mathcal{V}, \mathcal{E})$, where $\mathcal{V} = \{v_1, \ldots, v_n\}$ is the node set and $\mathcal{E} \subseteq \mathcal{V} \times \mathcal{V}$ is the edge set. For a node $u$, denote its neighbors as $\mathcal{N}(u) := \{v \in \mathcal{V} : \{u, v\} \in \mathcal{E}\}$. $G_u^K$ is the node-induced $K$-hop ego-network where the central node is $u$.

In the context of graph classification tasks, the input is typically represented as a set of graphs, where each graph $G_i$ is characterized by its node set $\mathcal{V}_i$, edge set $\mathcal{E}_i$, a node feature matrix $\mathbf{X}_i \in \mathbb{R}^{N_i \times D}$, and an adjacency matrix $\mathbf{A}_i \in \mathbb{R}^{N_i \times N_i}$. Here, $N_i$ is the total number of nodes in graph $G_i$, and $D$ is the dimensionality of the node features. The node feature matrix $\mathbf{X}_i$ represents the attributes of nodes in graph $G_i$. Each row, $\mathbf{x}_{i,v}$ corresponds to the $D$-dimensional feature vector of a node $v \in \mathcal{V}_i$. The adjacency matrix $\mathbf{A}_i$ describes the structure of the graph, where $\mathbf{A}_i[u, v] = 1$ if an edge $(u, v)$ exists in $\mathcal{E}_i$, and $\mathbf{A}_i[u, v] = 0$ otherwise. $G_{i,u}^K$ is the node-induced $K$-hop ego-network in graph $G_i$ where the central node is $u$, and $\mathbf{X}_i^{[u]}$ denotes the feature matrix for the involved nodes in $G_{i,u}^K$. The prediction targets for graph classification tasks are represented as $\mathbf{Y} \in \mathbb{R}^{N \times K}$, where $N$ is the number of graphs in the dataset, and $K$ is the number of classes. Each row $\mathbf{y}_i$ in $\mathbf{Y}$ is a one-hot vector representing the true class of graph $G_i$.

The entire dataset $\mathcal{D} = \{G_i, \mathbf{y}_i\}_{i=1}^N$ is divided into a training and validation set $\mathcal{D}_L = \{G_i, \mathbf{y}_i\}_{i=1}^{N_L}$ and a test set $\mathcal{D}_U = \{G_i\}_{i=N_L+1}^N$, where $N_L$ is the number of graphs in the training/validation set. In the training/validation phase, our goal is to learn a mapping function $\Phi : G_i \rightarrow \mathbf{y}_i, \forall i \in 1, \ldots, N_L$, using the labeled set $\mathcal{D}_L$. Once learned, the function $\Phi$ is expected to predict the true class labels of the unlabeled graphs in the test set $\mathcal{D}_U$.

### 2.2 Graph Neural Networks

In this paper, we focus on message-passing GNNs, where the representation $\mathbf{h}_v^{(l)}$ of each node $v$ in a graph $G$ is iteratively updated by aggregating information from its neighbors $\mathcal{N}(v)$. For the $l$-th layer, the updated representation is obtained via an AGGREGATE operation followed by an UPDATE operation:

$$\mathbf{m}_v^{(l)} = \text{AGGREGATE}^{(l)} \left( \left\{ \mathbf{h}_u^{(l-1)} : u \in \mathcal{N}(v) \right\} \right), \quad (1)$$

$$\mathbf{h}_v^{(l)} = \text{UPDATE}^{(l)} \left( \mathbf{h}_v^{(l-1)}, \mathbf{m}_v^{(l)} \right), \quad (2)$$

where $\mathbf{h}_v^{(0)} = \mathbf{x}_v$ is the initial node feature of node $v$ in graph $G$. For graph classification tasks, GNNs employ a READOUT function to aggregate the final layer node features $\left\{ \mathbf{h}_v^{(L)} : v \in \mathcal{V} \right\}$ into a graph-level representation $\mathbf{h}_G$:

$$\mathbf{h}_G = \text{READOUT} \left( \left\{ \mathbf{h}_v^{(L)} : v \in \mathcal{V} \right\} \right). \quad (3)$$

This graph-level representation is used for graph classification.

### 2.3 Graph Augmented Multi-Layer Perceptrons

GA-MLP (Graph-Augmented Multi-Layer Perceptrons) models [7] are a class of graph neural networks designed to understand graph

structure and enhance computational efficiency. These models operate in two primary steps: augmenting node features with linear operators based on the graph topology, and applying a node-wise learnable function. Formally, given a set of linear operators $\Omega = \{\omega_1(\mathbf{A}), \ldots, \omega_k(\mathbf{A})\} \subseteq \mathbb{R}^{|V| \times |V|}$, derived from the adjacency matrix $\mathbf{A}$, a GA-MLP first computes augmented features

$$\tilde{\mathbf{X}}_k = \omega_k(\mathbf{A}) \cdot \varphi(\mathbf{X}) \in \mathbb{R}^{n \times \tilde{d}}, \tag{4}$$

where $\varphi : \mathbb{R}^d \to \mathbb{R}^{\tilde{d}}$ is a learnable feature transformation, often realized by an MLP. The model then concatenates these features to get $\tilde{\mathbf{X}}$ and applies a learnable node-wise function $\rho$, to compute the final representation

$$Z = \rho(\tilde{\mathbf{X}}) \in \mathbb{R}^{n \times d'}. \tag{5}$$

A simplified version of GA-MLP takes $\varphi$ as the identity function, allowing pre-computation of the matrix products, thus improving computational efficiency. Various GA-MLP-type models, including SGC [48], GFN [8], gfNN [38], and SIGNs [15] have been proposed, showing competitive performances on diverse datasets.

## 3 RELATED WORK

Recently, Knowledge Distillation has proven to be effective for graph learning. Some previous works [14, 20, 27, 31, 41, 50, 53, 58, 62] have explored the distillation of knowledge from large teacher GNNs to smaller student GNNs. To further reduce the inference time and enable real-time applications, some recent works in this field explore GNN-to-MLP knowledge distillation. GLNN[61] adopts a soft logits-based KD method, which achieves predictive performance comparable to teacher GNN models, enabling real-time applications and significantly reducing inference time. KRD [51] explores the reliability of different knowledge points in GNNs and the diversity of roles they play in the distillation process. It introduces the KRD framework, which leverages reliable knowledge points to provide additional supervision signals. NOSMOG [45] introduces three key components: the incorporation of position features, representational similarity distillation, and adversarial feature augmentation to enhance the predictive performance of the student MLP compared to the vanilla soft logits-based KD method. FF-G2M [49] leverages both low-frequency and high-frequency components extracted from a single graph for full-frequency knowledge distillation. However, it is important to note that these methods are primarily designed for node classification and mainly operate on a single graph. It is not straightforward to adapt them to graph classification. In this work, we propose the first GNN-to-MLP KD framework for graph classification to bridge this gap.

## 4 PROPOSED FRAMEWORK

In this section, we present the details of MuGSI, which consists of three key components: graph-level distillation, subgraph-level distillation, and node-level distillation. The overall framework of MuGSI is illustrated in Figure 1.

### 4.1 Graph-Level Distillation

Currently, most GNN-to-MLP KD frameworks build upon response-based knowledge relying on the output of the last layer, i.e., soft logits. However, this approach fails to address the intermediate-level supervision from the teacher model, which turns out to be

important for representation learning using deep neural networks, as deep neural networks are good at learning multiple levels of feature representation with increasing abstraction [2]. Hence we resort to intermediate layers, i.e., feature maps as additional supervision signals, which serve as a good extension for soft logits-based KD approach. In MuGSI, we employ graph-level representation $h_G^T$ as a direct supervision signal from the teacher model for the student to emulate. This is because $h_G^T$ may encapsulate latent information that is concealed in soft logits. The *whole-graph distillation loss* can be formulated as follows:

$$\mathcal{L}_\mathcal{G} = \mathbb{E}_{G_i \sim \mathcal{D}_L} \left\| \frac{h_{G_i}^T}{\|h_{G_i}^T\|_2} - \frac{h_{G_i}^S}{\|h_{G_i}^S\|_2} \right\|_2^2, \tag{6}$$

where $h_{G_i}^T$ denotes the graph-level representation in the teacher model, and $h_{G_i}^S$ refers to the corresponding representation in the student model. The L2 norm, denoted by $\| \cdot \|_2$, measures the dissimilarity between these representations, thus driving the student to align with the teacher's graph-level representation.

### 4.2 Subgraph-Level Distillation

While whole-graph level representations provide meaningful learning signals for MuGSI, a more nuanced understanding of the structural information can be attained through subgraph-level distillation. Previous work in Computer Vision has found that the attention maps of hidden activations across image patches tend to have spatial correlations with predicted objects on the image level, and these correlations also tend to be higher in networks with higher accuracy [57].

In the context of graph-structured data, the concept of an image "patch" can be naturally analogized to a subgraph. This raises an important question: What type of subgraph should be selected as the underlying structure for a given graph? In this work, we elect to use clusters as the defining subgraphs, recognizing their essential role in understanding complex graph structures. For example, clusters in IMDB-BINARY [54] may correspond to groups of actors who frequently co-star in the same films. In REDDIT-BINARY [54], clustering nodes (users) can reveal community structures or groups of users that interact more frequently with each other. This could reflect shared opinions, interests, or other social dynamics within that specific thread. Although for some other scenarios, such as bioinformatics, clusters do not necessarily have a straightforward interpretation as they might be in social networks, they could be used to identify structural motifs or common substructures within a molecule, depending on the features used. This suggests that clusters as graph "patches" can provide valuable information for graph classification.

In MuGSI, we maximize the inter-cluster similarity by leveraging the kernel matrix $\mathbf{K}$, which embodies pairwise interactions among clusters and allows for describing the geometry of the corresponding feature spaces [23]. Specifically, given two clusters $i$ and $j$, let $C_i$ and $C_j$ denotes the node sets belonging to cluster $i$ and $j$ respectively, then we can calculate a subgraph-level representation $\mathbf{h}_{C_i}$ and $\mathbf{h}_{C_j}$ similarly in Eq. 3, i.e., $\mathbf{h}_{C_i} = \text{READOUT}\left(\left\{\mathbf{h}_v^{(L)} : v \in C_i\right\}\right)$. A kernel matrix $\mathbf{K} \in R^{N_C \times N_C}$ is obtained where $N_C$ denotes the number of clusters in the given graph $G$, and each element $k_{ij} =$

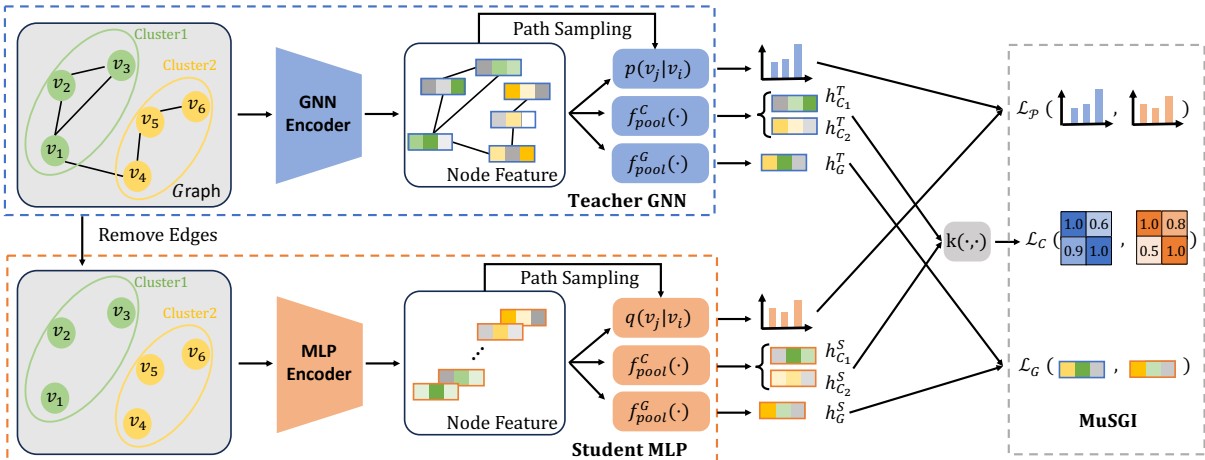

**Figure 1: The figure illustrates the KD process with multi-granularity distillation loss. First a teacher GNN model is pre-trained, then an MLP-type student model is trained using the distilled multi-granularity structural knowledge from the teacher model: (a) whole-graph distillation loss $\mathcal{L}_{\mathcal{G}}$; (b) inter-cluster distillation loss $\mathcal{L}_C$; (c) path-consistency loss $\mathcal{L}_{\mathcal{P}}$. Note that the soft logits distillation loss $\mathcal{L}_{SL}$ and the ground-truth cross-entropy loss $\mathcal{L}_{GT}$ are not shown in the figure. $f_{pool}^G(\cdot)$ and $f_{pool}^C(\cdot)$ denote the graph pooling and cluster pooling function for teacher GNNs and student MLPs. In MuSGI, the graph pooling function and cluster pooling function share the same function form, e.g., *summation* or *attention*. If attention-based pooling is utilized, graph pooling and cluster pooling share the same parameters to improve generalizability.**

$k(\mathbf{h}_{C_i}, \mathbf{h}_{C_j})$. $k(\cdot, \cdot)$ is a kernel function that projects the sample vectors into a higher or infinite dimensional feature space. We use Cosine Similarity as the kernel function, i.e.,

$$k_{ij} = k(\mathbf{h}_{C_i}, \mathbf{h}_{C_j}) = \frac{\langle \mathbf{h}_{C_i}, \mathbf{h}_{C_j} \rangle}{\|\mathbf{h}_{C_i}\|_2 \cdot \|\mathbf{h}_{C_j}\|_2}. \tag{7}$$

We then define the *inter-cluster distillation loss* $\mathcal{L}_C$ as following:
$$\mathcal{L}_C = \|\mathbf{K}_S - \mathbf{K}_T\|_F^2. \tag{8}$$

Here $\mathbf{K}_S$ and $\mathbf{K}_T$ are the obtained kernel matrix from the student and teacher model respectively, i.e., $\mathbf{K}_S[i, j] = k(\mathbf{h}_{C_i}^S, \mathbf{h}_{C_j}^S)$, and $\mathbf{K}_T[i, j] = k(\mathbf{h}_{C_i}^T, \mathbf{h}_{C_j}^T)$. Here $\mathbf{h}_{C_i}^S$ and $\mathbf{h}_{C_i}^T$ are the cluster-level representation obtained for cluster $i$ from the teacher model and student model respectively.

## 4.3 Node-Level Distillation

Graph neural network's success in graph classification is closely related to the Weisfeiler-Lehman (1-WL) algorithm. By iteratively aggregating neighboring node features to a center node, both 1-WL and GNN obtain a node representation that encodes a rooted subtree around the center node. These rooted subtree representations are then pooled into a single representation to represent the whole graph [60].

Hence, to obtain a discriminative representation for the whole graph, it is necessary to learn a "good" representation for each node $v$ that captures its local substructure. Let $\mathbf{H}_T = f_T(\mathbf{X}, \mathbf{A})$ and $\mathbf{H}_S = f_S(\mathbf{X}, (\mathbf{A}))$ denote the node representations for a given graph $G$ obtained from teacher model and student model respectively (($\mathbf{A}$) means $\mathbf{A}$ is optional depending on the choice of $f_S$), $\mathbf{h}_v^T$ and $\mathbf{h}_v^S$ denote the representation for node $v$ from the teacher and student model. If we assume $f_T(\cdot)$ is more expressive than $f_S(\cdot)$, then $\mathbf{h}_v^T$ should more accurately reflect the local substructure of

node $v$ compared to $\mathbf{h}_v^S$. As the local substructure of every node $v \in \mathcal{V}$ is essential for graph classification, we propose a novel node-level component in MuSGI to transfer the local structural knowledge from the teacher to the student model. This is done by maximizing the agreement between the teacher and student model on their opinions regarding the similarity of local neighborhood nodes. Specifically, for each node $v$, let $\mathcal{P}_v^K$ be a collection of $K$-step random-walk paths starting from node $v$, a single path drawn from $\mathcal{P}_v^K$ is denoted as $p_v := (p_v^1, \cdots, p_v^K)$. To measure the similarity between node $v$ and its neighboring nodes along the random-walk path $p_v$, we define the following conditional probability $p(u \mid v)$ for the teacher model:

$$p(u \mid v) = \frac{e^{h_u^T h_v}}{\sum\limits_{w \in p_v} e^{h_w^T h_v}}, u \in p_v, \tag{9}$$

and $q(u \mid v)$ is similarly defined for the student model. The *path consistency distillation loss* is defined as follows:

$$\mathcal{L}_{\mathcal{P}} = \mathbb{E}_{v \sim \mathcal{V}} \mathbb{E}_{p_v \sim \mathcal{P}_v^K} \mathcal{D}_{KL}(p(u \mid v), q(u \mid v)) \tag{10}$$

**Node feature augmentation.** In addition to the node-level distillation of local substructures, the bottleneck of expressiveness of student models still needs to be addressed. As discussed in Section 1, the input feature space is typically very small for graph classification datasets, which severely limits the expressive power of the student model, and its learning capability, hence in MuSGI we enhance the node features by incorporating structure-aware features. Specifically, we utilize Laplacian eigenvectors [1] as node positional encoding which is shown to be effective across various message-passing GNNs [10]. To further address this issue, we propose using a 1-hop GA-MLP as a more expressive student model.

Notably, an MLP is essentially a 0-hop GA-MLP. Although the expressive power of a GA-MLP is still exponentially lower than a GNN model in terms of the number of equivalence classes [7], the student model can achieve comparable or superior results to the teacher GNN when combined with MuGSI for knowledge transfer.

**Overall Framework.** The final objective $\mathcal{L}$ of the proposed framework MuGSI is defined as a weighted combination of ground-truth cross-entropy loss $\mathcal{L}_{GT}$, soft logits distillation loss $\mathcal{L}_{SL}$, and the multi-granularity distillation loss $\mathcal{L}_{\mathcal{G}}$, $\mathcal{L}_{C}$ and $\mathcal{L}_{\mathcal{P}}$ respectively.

$$\mathcal{L} = \mathcal{L}_{GT} + \mathcal{L}_{SL} + \lambda \mathcal{L}_{\mathcal{G}} + \mu \mathcal{L}_{C} + \eta \mathcal{L}_{\mathcal{P}}, \qquad (11)$$

where $\lambda$, $\mu$ and $\eta$ are trade-off weights for balancing $\mathcal{L}_{\mathcal{G}}$, $\mathcal{L}_{C}$ and $\mathcal{L}_{\mathcal{P}}$, respectively. For $\mathcal{L}_{SL}$, the weight is set to 1.0 without any hyper-parameter tuning for MuGSI.

## 5 NUMERICAL EXPERIMENTS

In this section, we extensively evaluate the effectiveness, efficiency, and robustness of the proposed framework MuGSI by investigating the following research questions.

**RQ1**: We start with a vanilla MLP (with LaPE) as the student model to evaluate MuGSI's performance, and ask: *How does MuGSI perform for student MLPs?* **RQ2**: We further adopt a GA-MLP (with LaPE) as the student model, to verify whether the KD process will be more effective with a more capable learner, and ask: *How does MuGSI perform for more expressive student architecture?* **RQ3**: As MuGSI is model-agnostic, and can be combined with different teacher model architectures, we then evaluate MuGSI's performance with two different GNN architectures with varied representative power. We ask: *How does MuGSI perform for different teachers?* **RQ4**: Given the dynamic nature of many real-world graphs, we evaluate MuGSI's robustness and efficiency in such environments, and ask: *How robust and efficient is MuGSI in dynamic environments?* **RQ5**: As multi-granularity loss consists of three key components, we study how each independent component contributes to the KD process, and ask: *How does each component perform in MuGSI?* **RQ6**: Finally, we study the impact of hyper-parameters in MuGSI, and ask: *How do different hyper-parameters affect the performance of MuGSI?*

### 5.1 Experiment Settings

**Datasets**. We use 6 small real-world datasets and 2 large real-world datasets to evaluate our proposed framework. For the 6 small real-world datasets from TUDataset [36], PROTEINS [9],NCI1 [47], BZR [43] and DD [9, 42] are bioinformatics datasets; REDDIT-BINARY and IMDB-BINARY are social network datasets. As no node features are provided for the social network datasets, we use one-hot encoding of node degrees as their node features. For the 2 large real-world datasets, we use CIFAR10 from Benchmarking GNNs [10], and MolHIV from Open Graph Benchmark [24]. See Appendix A.5 for the dataset statistics.

**Model Architectures**. As a model-agnostic framework, MuGSI can be combined with any teacher GNN architecture. In this work, we adopt three GNN teacher model architectures: GIN [52], GCN [29] and KPGNN [13]. For student model architectures, MLP and GA-MLP are both adopted to thoroughly evaluate MuGSI's performance with students of different expressiveness levels. For GA-MLP,

a simplified version is utilized with 1-hop neighborhood aggregation, i.e., $\Omega = \left\{ \mathbf{I}, \mathbf{AD}^{-1} \right\}$ and $\phi$ being the identity function, using the notation from Section 2.3. This simplified version allows pre-computation, leading to the time complexity of this GA-MLP architecture becoming close to that of a standard MLP.

**Baselines**. We consider several baseline methods to facilitate a comprehensive evaluation of our proposed framework. **MLP**: We use MLP as the basis for comparison with more advanced methods. **GLNN$_{MLP}$**: This method distills student MLPs using soft labels, which is similar to GLNN [61], except that a graph pooling function is utilized to obtain a graph-level representation. **MLP+LaPE**: Here, we extend the MLP by augmenting it with node features encoded through Laplacian eigenvector positional encodings (LaPE). This enhancement aims to increase the expressiveness of the student MLP model. **GLNN$_{MLP+LaPE}$**: This method combines the MLP with both Laplacian eigenvector positional encodings (LaPE) and soft logits-based KD, serving as a more advanced variant of GLNN. We extend the same experiment setting for GA-MLP, specifically, the baseline methods are **GA-MLP**, **GLNN$_{GA-MLP}$**, **GA-MLP+LaPE**, and **GLNN$_{GA-MLP+LaPE}$**. **NOSMOG**: We also adopt NOSMOG [45] as another strong baseline method for comparison. As Deep-Walk [39] generates node embeddings in a transductive manner, which is not suitable for graph classification, we use LaPE to replace this component in NOSMOG. We use **NOSMOG$_{MLP}$** and **NOSMOG$_{GA-MLP}$** to denote NOSMOG applied to student MLP and GA-MLP respectively, also note that LaPE is an inherent component in NOSMOG, which injects structural features to student models. Finally, we denote **MLP*** as the best performing model between MLP and MLP+LaPE, similarly for **GA-MLP***.

**Evaluation Protocol**. For the 6 real-world datasets from TU-Dataset, we use the standard stratified splits [52], and perform 10-fold cross-validation with 90% training and 10% testing, we report the mean best test results. The teacher GNN model for each fold is saved based on the best test result and, hence is consistent with the reported test results from student models. For CIFAR10, we use standard split that consists of 45,000 train, 5,000 validation, and 10,000 test graphs, we report the test classification accuracy according to the best validation accuracy. For MolHIV, we follow the scaffold split[25, 40], the split for train/validation/test sets is 80%:10%:10%. We report the ROC-AUC value on the test set according to the best ROC-AUC on the validation set.

### 5.2 How Does MuGSI Perform for Student MLPs? (RQ1)

We first evaluate MuGSI where the student models are MLPs, and compare with MLP-related baseline methods. The experimental results are illustrated in Table 2, from which we can make several observations: (1) Incorporating Laplacian eigenvectors into the vanilla MLP models is able to enhance their classification performance across various datasets. Notable improvements include an increase of 4.06% in PROTEINS, 7.96% in DD, and 4.81% in REDDIT-BINARY. (2) While the use of soft logits[22] has shown significant benefits for node classification, as evidenced by [61], its impact on graph classification is negligible for most datasets. This finding aligns with our analysis. (3) Our proposed MuGSI$_{MLP*}$ framework consistently outperforms other variations such as GLNN$_{MLP}$ and

|  | PROTEINS | BZR | DD | NCI1 | IMDB-B | REDDIT-B | CIFAR10 | MolHIV |
|---|---|---|---|---|---|---|---|---|
| GIN | 79.25±3.22 | 93.09±1.89 | 77.67±2.86 | 82.43±1.12 | 79.60±3.02 | 91.35±1.58 | 55.57 | 76.43 |
| MLP | 72.61±2.98 | 79.26±1.50 | 73.59±2.90 | 59.56±1.46 | 77.11±2.76 | 80.81±2.36 | 51.57±0.19 | 65.31±1.49 |
| MLP+LaPE | 75.92±2.63 | 81.73±2.21 | 79.45±2.79 | 66.05±2.01 | 76.60±2.61 | 84.70±2.44 | 48.34±0.08 | 64.72±1.07 |
| $GLNN_{MLP}$ | 72.96±2.54 | 79.51±1.94 | 74.49±2.94 | 59.95±2.33 | 77.58±3.27 | 80.21±2.60 | 51.61±0.26 | 68.38±1.01 |
| $GLNN_{MLP+LaPE}$ | 76.74±4.50 | 82.47±2.38 | 79.36±3.03 | 66.93±1.32 | 77.59±3.27 | 84.86±3.97 | 49.34±0.34 | 67.56±0.52 |
| $NOSMOG_{MLP}$ | 76.71±3.79 | 84.41±4.46 | 79.96±3.04 | 68.29±2.07 | 77.02±4.43 | 84.61±2.78 | 48.49±0.31 | 64.56±1.76 |
| $MuGSI_{MLP^*}$ (ours) | 77.1±3.59 | 85.68±2.26 | 80.33±2.76 | 67.71±2.43 | 78.06±3.02 | 87.91±1.37 | 51.89±0.21 | 71.92±0.71 |
| $\Delta_{MLP}$ | 4.49(6.18%) | 6.42(8.10%) | 6.74(9.16%) | 8.15(13.68%) | 0.95(1.23%) | 7.10(8.79%) | 0.32(0.62%) | 6.61(10.12%) |
| $\Delta_{GLNN_{MLP^*}}$ | 0.36(0.47%) | 3.21(3.89%) | 0.97(1.22%) | 0.78(1.17%) | 0.58(0.75%) | 3.05(3.59%) | 0.28(0.54%) | 3.54(5.18%) |
| $\Delta_{NOSMOG_{MLP}}$ | 0.39(0.50%) | 1.27(1.48%) | 0.37(0.46%) | -0.58(-0.85%) | 1.04(1.33%) | 3.30(3.75%) | 3.39(6.55%) | 7.36(10.23%) |
| $\Delta_{GIN}$ | -2.15(-2.71%) | -7.41(-7.96%) | 2.66(3.42%) | -14.72(-17.86%) | -1.54(-1.93%) | -3.44(-3.77%) | -3.68(-6.62%) | -4.51(-5.90%) |
| GA-MLP | 75.74±2.68 | 90.62±3.81 | 75.71±1.73 | 75.93±1.98 | 79.95±3.02 | 88.45±2.36 | 54.81±0.19 | 71.55±1.08 |
| GA-MLP+LaPE | 75.47±2.58 | 87.42±3.67 | 78.85±2.49 | 71.61±2.02 | 78.45±3.26 | 89.62±2.86 | 51.91±0.18 | 71.78±1.48 |
| $GLNN_{GA-MLP}$ | 75.76±3.41 | 91.97±3.92 | 76.73±2.52 | 75.45±2.28 | 80.20±3.19 | 88.07±1.83 | 54.88±0.26 | 73.74±0.92 |
| $GLNN_{GA-MLP+LaPE}$ | 76.28±2.61 | 87.39±2.79 | 79.86±2.31 | 71.94±2.14 | 79.40±3.92 | 89.55±2.21 | 52.85±0.27 | 72.91±0.86 |
| $NOSMOG_{GA-MLP}$ | 78.35±2.74 | 88.78±2.32 | 80.41±3.57 | 74.84±2.92 | 79.10±3.72 | 89.09±1.64 | 51.36±0.46 | 73.67±1.31 |
| $MuGSI_{GA-MLP^*}$ (ours) | 78.26±4.78 | 93.1±2.81 | 81.57±2.24 | 76.86±2.33 | 80.31±3.36 | 90.91±2.05 | 55.63±0.31 | 76.38±0.95 |
| $\Delta_{GA-MLP}$ | 2.52(3.33%) | 2.48(2.74%) | 5.86(7.74%) | 0.93(1.22%) | 0.36(0.45%) | 2.46(2.78%) | 0.82(1.50%) | 4.83(6.75%) |
| $\Delta_{GLNN_{GA-MLP^*}}$ | 1.98(2.60%) | 1.13(1.23%) | 1.71(2.14%) | 1.41(1.87%) | 0.11(0.14%) | 1.36(1.52%) | 0.75(1.37%) | 2.64(3.58%) |
| $\Delta_{NOSMOG_{GA-MLP}}$ | -0.08(-0.11%) | 4.32(4.64%) | 1.16(1.42%) | 2.02(2.63%) | 1.21(1.50%) | 1.82(2.01%) | 4.27(7.68%) | 2.71(3.68%) |
| $\Delta_{GIN}$ | -0.99(-1.25%) | 0.01(0.01%) | 3.90(5.02%) | -5.57(-6.76%) | 0.71(0.89%) | -0.43(-0.48%) | 0.06(0.11%) | -0.05(-0.07%) |

**Table 2: Experiment results where the teacher model is GIN, and the student models are MLP, MLP+LaPE and GA-MLP, GA-MLP+LaPE. The absolute improvement and relative improvement are both illustrated in the table. As illustrated in the figure, MuGSI outperforms other competitive baseline methods on almost all the datasets, with different student MLP-type model architectures. Using GA-MLP as the student model, MuGSI exhibits comparable performance with the teacher GIN model in 7/8 datasets.**

| Teacher | Student | PROTEINS | IMDB-BINARY | DD | BZR |
|---|---|---|---|---|---|
| - | GA-MLP | 75.74±2.68 | 79.95±3.02 | 75.71±1.73 | 90.62±3.81 |
| - | GA-MLP+LaPE | 75.47±2.58 | 78.45±3.26 | 78.85±2.49 | 89.62±2.86 |
| GCN | - | 76.28±2.71 | 79.27±4.16 | 76.31±1.44 | 89.88±3.38 |
|  | $GLNN_{GA-MLP}$ | 75.57±2.73 | 80.01±4.05 | 75.40±3.09 | 92.08±2.52 |
|  | $GLNN_{GA-MLP+LaPE}$ | 77.09±2.83 | 79.60±3.13 | 79.62±2.12 | 88.68±3.66 |
|  | $MuGSI_{GA-MLP^*}$ | 77.69±2.67 | 81.09±3.91 | 80.52±2.29 | 91.94±3.07 |
|  | $\Delta_{GA-MLP}$ | 1.95(2.57%) | 1.14(1.43%) | 4.81(6.35%) | 1.32(1.46%) |
|  | $\Delta_{GLNN_{GA-MLP^*}}$ | 0.59(0.77%) | 1.07(1.35%) | 0.89(1.13%) | -0.14(-0.15%) |
|  | $\Delta_{GCN}$ | 1.41(1.85%) | 1.82(2.30%) | 4.21(5.52%) | 2.06(2.29%) |
| KPGIN | - | 78.56±3.17 | 80.30±4.37 | 81.07±2.83 | 93.11±2.51 |
|  | $GLNN_{GA-MLP}$ | 76.01±2.56 | 80.50±4.01 | 76.14±3.29 | 91.72±2.31 |
|  | $GLNN_{GA-MLP+LaPE}$ | 76.37±3.84 | 79.80±2.84 | 81.23±3.57 | 89.17±3.99 |
|  | $MuGSI_{GA-MLP^*}$ | 77.13±2.53 | 81.04±3.82 | 82.64±3.31 | 92.89±3.54 |
|  | $\Delta_{GA-MLP}$ | 1.39(1.84%) | 1.09(1.36%) | 6.93(9.15%) | 2.27(2.50%) |
|  | $\Delta_{GLNN_{GA-MLP^*}}$ | 0.75(1.00%) | 0.54(0.67%) | 1.40(1.74%) | 1.17(1.28%) |
|  | $\Delta_{KPGIN}$ | -1.43(-1.82%) | 0.74(0.92%) | 1.57(1.94%) | -0.22(-0.24%) |

**Table 3: Experiment results with different teacher GNN model architectures, the student model is GA-MLP\*. Notably, even with a 3-WL equivalent model architecture KPGIN as the teacher model, MuGSI is able to achieve comparable or superior performance using 1-hop GA-MLP as the student model. This demonstrates the effectiveness of MuGSI.**

$GLNN_{MLP+LaPE}$ across different datasets. Notably, MuGSI outperforms $GLNN_{MLP^*}$ by 3.89% in BZR, 3.59% in REDDIT-BINARY, and 6.45% in MolHIV, highlighting the effectiveness of our framework for graph classification tasks. (4) NOSMOG also excels across several datasets, thanks to its representational similarity distillation component, which aligns the distribution over local substructures between the teacher GNNs and the student MLPs, since it shares the same function form with inter-cluster distillation loss $\mathcal{L}_C$, however this component works at the node level, and incurs a high space complexity of $O(|\mathcal{V}|^2)$. In contrast, the path-consistency distillation loss $\mathcal{L}_P$ is more memory efficient ($O(1)$ space complexity since

the random-walk path length is a fixed constant). Furthermore, the multi-granularity structural distillation introduced in MuGSI generally outperforms NOSMOG, which solely relies on node-level structural distillation. (5) However, we also notice that the improvements are slight in several datasets. We hypothesize that this could be attributed to the limited expressive power of the student model architecture and the constraints imposed by the small size of the input feature space. These results lead us to consider an intriguing question: what might be achieved with a more expressive student model?

### 5.3 How Does MuGSI Perform for More Expressive Student Architecture? (RQ2)

We adopt a 1-hop GA-MLP as the student model in this experiment. The experiment results are illustrated in Table 2, from which we can make several observations: (1) $MuGSI_{GA-MLP^*}$ achieves the best performance across 7/8 datasets, demonstrating its effectiveness. (2) For several datasets, the enhanced expressiveness of the student model yields a larger performance gain over $GLNN_{MLP^*}$, e.g., 0.47% versus 2.59% in PROTEINS and 1.22% versus 2.15% in DD, suggesting that a more expressive learner can sometimes be a "smarter" learner. (3) For several datasets such as DD and IMDB-BINARY, using GA-MLP on its own without the aid of knowledge distillation already achieves comparable or even superior performance compared with the teacher GIN model. Nevertheless, utilizing MuGSI further enhances the student model's performance. (4) When adopting GA-MLP\* as the student model, $MuGSI_{GA-MLP^*}$ exhibits performance on par with the teacher model in 7/8 datasets and surpasses the teacher model in 4/8 datasets. This shows the effectiveness of our proposed knowledge distillation framework.

| Datasets | MLP* | w/ GraphKD | w/ ClusterKD | w/ NodeKD | MuGSI$_{MLP*}$ | $\Delta_{GraphKD}$ | $\Delta_{ClusterKD}$ | $\Delta_{NodeKD}$ | $\Delta_{MuGSI}$ |
|---|---|---|---|---|---|---|---|---|---|
| PROTEINS | 75.92±2.63 | 76.28±4.31 | 76.73±3.71 | 76.49±3.33 | **77.10±3.59** | 0.36(0.47%) | 0.81(1.07%) | 0.57(0.75%) | 1.18(1.55%) |
| BZR | 81.73±2.21 | 84.20±3.48 | 84.68±3.24 | 83.71±3.46 | **85.68±2.26** | 2.47(3.02%) | 2.95(3.61%) | 1.98(2.42%) | 3.95(4.83%) |
| DD | 79.45±2.79 | 79.87±2.95 | 80.13±3.25 | 79.96±2.53 | **80.33±2.76** | 0.42(0.53%) | 0.68(0.86%) | 0.51(0.64%) | 0.88(1.11%) |
| REDDIT-BINARY | 84.70±2.44 | 85.85±2.31 | 86.73±2.28 | 86.11±1.97 | **87.91±1.37** | 1.15(1.36%) | 2.03(2.4%) | 1.41(1.66%) | 3.21(3.79%) |

**Table 4: Ablation study for independent components in MuGSI$_{MLP*}$, in which the teacher model is GIN. As shown in the table, each independent component in MuGSI makes a positive contribution to the KD process.**

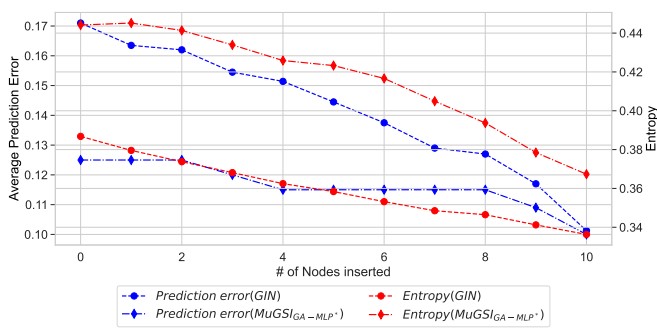

**Figure 2: Average prediction error and entropy resulted by GIN and MuGSI$_{GA-MLP*}$ when sequentially inserting 10 nodes back to the graphs. As demonstrated, MuGSI$_{GA-MLP*}$ is more robust and less susceptible to topological changes.**

## 5.4 How Does MuGSI Perform for Different Teachers? (RQ3)

As the expressive power of GIN is upper bounded by 1-WL[37, 52], recently there is a collection of literature proposed to enhance the expressivity of message-passing GNNs. To explore how different teacher model architectures with different levels of expressiveness affect the knowledge distillation process, we adopt another two teacher model architectures: GCN [29] and KPGIN [13]. The expressive power of GCN is also upper bounded by 1-WL, and KPGIN is a $K$-hop message-passing GNN model with peripheral subgraph information, which is strictly more powerful than 1-WL and is upper bounded by 3-WL. We adopt GA-MLP and GA-MLP+LaPE as the student models.

As illustrated in Table 3, we can see that (1) For both GCN and KP-GIN as teacher models, MuGSI$_{GA-MLP*}$ outperform GLNN$_{GA-MLP*}$ in most datasets, which demonstrates that MuGSI as a model-agnostic KD framework is effective. (2) The performance of vanilla GA-MLP* is on par with GCN or even superior to GCN, e.g., in DD and IMDB-BINARY. However, distilling knowledge from GCN into GA-MLP* using MuGSI can still benefit student GA-MLP* significantly. For instance, the accuracy for GA-MLP* improves from 78.85% to 80.52% using MuGSI in the DD dataset although GCN merely achieves 76.31% accuracy, similarly in IMDB-BINARY, the classification accuracy of GA-MLP* improves from 79.95% to 81.09% using MuGSI, while GCN achieves 79.27%. Furthermore, MuGSI$_{GA-MLP*}$ outperforms GCN in all 4 datasets with a large margin. (3) For a more powerful teacher model KPGIN, MuGSI$_{GA-MLP*}$ also consistently outperforms GLNN$_{GA-MLP*}$. Notably, even if KP-GIN is 3-WL equivalent, a 1-hop student GA-MLP* using MuGSI achieves comparable or superior performance.

## 5.5 How Robust and Efficient is MuGSI in Dynamic Environments? (RQ4)

In practical production environments, graphs are often dynamic, with nodes being inserted or removed over time. Taking REDDIT-BINARY as an example, each node in a graph corresponds to a user engaged in a discussion thread, and edges represent interactions between these users. As nodes can be added or removed, this may lead to distributional shift issues. In this section, we verify how does teacher GIN model and student GA-MLP* perform under this scenario. We utilize the first fold of the REDDIT-BINARY dataset for our experiments. For each graph $G$ in the test set, we first randomly remove 10 nodes from the graph, then we insert them back sequentially to get the same graph $G$. This process is repeated 20 times for each graph in the test set. As we only remove a small fraction of nodes (2%-3% at most) in each graph, it is reasonable to assume that the graph's label remains unchanged. We calculate two metrics: (1) *Average prediction error*. For each perturbed graph with $k$ inserted nodes where $k \in [0, 10]$, we assess whether its predicted label matches that of the original graph. The error is binary: 0 for a match and 1 otherwise, and we calculate the average error across all perturbations for each $k$. (2) *Average entropy*. Instead of recording a binary variable, we compute the Shannon entropy of the predicted label distribution for each perturbed graph with $k$ insertions. For incorrect predictions, we set the entropy to its maximum value (i.e., 1.0 for binary classification). This metric helps to quantify the confidence of the model predictions.

The average prediction error for MuGSI$_{GA-MLP*}$ is significantly lower than that for GIN, as depicted in Figure 2, despite comparable accuracies on unperturbed test graphs (90.1% for MuGSI$_{GA-MLP*}$ vs. 89.86% for GIN). Specifically, GIN's accuracy drops by 7.76% upon the removal of 10 nodes, while MuGSI$_{GA-MLP*}$'s accuracy decreases only by 2.77%, this demonstrates the robustness of the student model. We hypothesize that the robustness of MuGSI$_{GA-MLP*}$ arises from the structural information retained in the model parameters during the knowledge distillation process, which is orthogonal to topological changes. Additionally, the receptive field of GIN (5 hops in this case) is much larger than a 1-hop GA-MLP, hence is more susceptible to the topological changes. Despite its higher average prediction error, GIN's model predictions exhibit greater confidence (i.e., lower entropy) compared to those from MuGSI$_{GA-MLP*}$.

Regarding efficiency, as illustrated in Figure 3, GA-MLP* is substantially faster than GIN. This is due to that it takes the entire input **A** and **X** to re-calculate the model prediction for GIN; whereas for GA-MLP*, with *sum* pooling as readout function, we can obtain a static representation first given the graph with 10 nodes removed, then for each node inserted back, we only need to incrementally calculate its representation and sum it with the static representation, followed by a linear transformation. The static representation

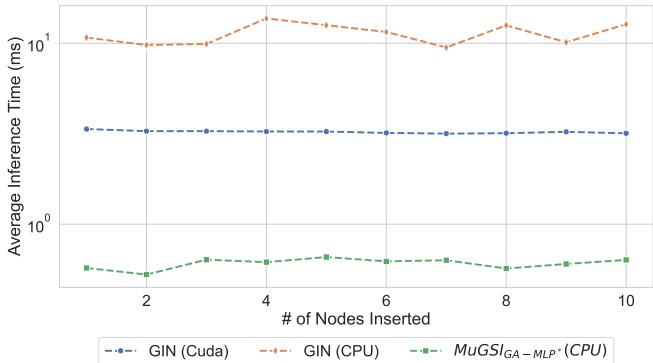

**Figure 3: Average inference time from GIN and MuGSI$_{GA-MLP^*}$ when sequentially inserting 10 nodes back to the graphs. MuGSI$_{GA-MLP^*}$ is 17.18x faster than GIN during the inference stage using a CPU machine.**

can also be updated with one additional operation. This procedure significantly reduces computational overhead, allowing GA-MLP* to achieve an average incremental inference time of **0.59ms** on a CPU machine, which is **17.18x** faster than GIN using a CPU machine and 4.98x faster than GIN using a CUDA machine. The efficiency makes the student model deployable in resource-constrained environments.

## 5.6 How Does Each Component Perform in MuGSI? (RQ5)

As MuGSI consists of three components for multi-scaled structural knowledge distillation, we explore how each independent component affects the KD process for several datasets. To ensure a fair comparison, MLP* is adopted as the baseline method, since we use MLP* as the student model for knowledge distillation. The three components are named GraphKD, ClusterKD, and NodeKD as illustrated in Table 4. We can see that: (1) Each independent component makes a positive contribution to the KD process; (2) For the 4 datasets, ClusterKD consistently brings the largest performance gain; (3) MuGSI leverages joint structural knowledge distillation, outperforms the individual components, showcasing the effectiveness of distilling multi-granularity structural information.

## 5.7 How Do Different Hyper-parameters Affect the Performance of MuGSI? (RQ6)

We first provide sensitivity analysis for $\lambda$, $\mu$, and $\eta$, which control the strength of each distillation component in MuGSI. We perform grid search on $\lambda \in (1.0, 1e-1, 1e-2)$, $\mu \in (1.0, 1e-1, 1e-2)$, $\eta \in (1e-4, 1e-5)$, leading to 18 models with different hyper-parameter combinations. We index these models from 0 to 17, as illustrated in Figure 4. As we can see, the correlation for MuGSI$_{MLP^*}$ and MuGSI$_{GA-MLP^*}$ in REDDIT-BINARY is much higher than that in the BZR dataset, possibly because REDDIT-BINARY is a much larger dataset than BZR (2000 samples vs. 405 samples); Furthermore, MLP* and GA-MLP* both utilize Laplacian eigenvectors in REDDIT-BINARY, whereas in BZR, MLP* is MLP+LaPE, and GA-MLP* is GA-MLP. This may lead to different inductive biases for MuGSI$_{MLP^*}$ and MuGSI$_{GA-MLP^*}$ in BZR, leading to lower correlation between two different student models with different hyper-parameters.

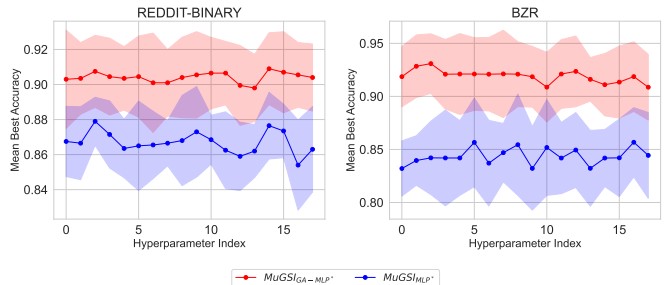

**Figure 4: Mean best accuracy for different hyper-parameter combinations for MuGSI$_{MLP^*}$ and MuGSI$_{GA-MLP^*}$.**

|  | PROTEINS | DD | BZR | IMDB-BINARY |
|---|---|---|---|---|
| $\mathcal{L}_4$ | 77.54±2.92 | 81.15±1.93 | 92.63±3.68 | **81.03±3.26** |
| $\mathcal{L}_8$ | **78.26±4.78** | **81.57±2.24** | **93.10±2.81** | 80.31±3.36 |
| $\mathcal{L}_{12}$ | 77.89±3.45 | 80.98±2.08 | 92.80±3.64 | 80.81±3.25 |
| $\mathcal{L}_{16}$ | 78.05±3.84 | 80.31±2.33 | 92.66±2.46 | 80.20±3.96 |

**Table 5: Analysis of the effect of random walk path lengths on MuGSI$_{GA-MLP^*}$ with GIN as the teacher model. Here, $\mathcal{L}_i$ denotes a random walk path of length $i$.**

The random-walk path length is another key hyper-parameter in the path consistency loss $\mathcal{L}_\mathcal{P}$. We do an ablation study and investigate the impact of various random-walk path lengths, from which several observations can be made: *(1)* The choice of an optimal random walk path length appears to be influenced by the inherent topological structure of graphs within specific datasets. This suggests that the most effective path length is not universally constant, but rather is subject to the unique characteristics of each dataset. *(2)* Extended random walk path lengths generally yield sub-optimal results. One possible explanation for this trend is that longer paths could introduce additional noise during the knowledge distillation process in capturing local substructures.

## 6 CONCLUSION

In this paper, we identified an under-explored problem: the GNN-to-MLP distillation for graph classification, then we offer an analysis of why existing GNN-to-MLP KD frameworks are suboptimal for graph classification. We then introduce MuGSI, the first GNN-to-MLP Knowledge Distillation framework for graph classification. This framework incorporates a node feature augmentation component to enhance the expressiveness of student MLPs and make them more capable learners; MuGSI also proposes a novel multi-granularity distillation loss to generate dense learning feedback and facilitate comprehensive knowledge transfer from the teacher model to the student model. MuGSI is model-agnostic, demonstrating comparable performance across a variety of teacher model architectures including KPGNN, a 3-WL equivalent model architecture using 1-hop GA-MLP as the student model. Moreover, MuGSI is robust and efficient in dynamic environments, which serves as a potent technique to tackle test-time distribution shift issues, and enables fast inference in environments with limited computational resources.

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

# A APPENDIX

## A.1 More Experiment Setting and Implementation Details

**Teacher GNN Models.** The hyper-parameter search spaces for GCN and GIN include: the number of layers $L = \{2, 3, 5\}$, dropout rate $p = \{0, 0.5\}$. The hidden size $H = \{32, 64\}$. The search space for KPGIN is: number of layers $L = \{2, 3, 4\}$, dropout rate $p = \{0, 0.5\}$, number of hops $K = \{3, 4\}$, combine function $F = \{attention, geometric\}$. The kernel is the shortest path kernel and the hidden size is 64 for all the datasets. For TUDataset, the GNN model is selected based on the mean best validation accuracy. For CIFAR10 the teacher model is obtained according to the best accuracy on the test dataset, and for MolHIV, the teacher GNN model is obtained according to the best ROCAUC value on the test dataset.

**Student MLP Models.** The hyper-parameter search space for MLP and GA-MLP is: number of layers $L = \{3, 4\}$, dropout is not used for TUDataset and CIFAR10; for MolHIV, dropout is set to 0.5. The hidden size of student models is set to 64 uniformly.

**Model Training.** For TUDataset, all models are trained for 350 epochs, initial learning rate is $8e - 3$, with a decaying factor of 0.6 with patience to be 30 epochs. For CIFAR10, all models are trained for 120 epochs. The initial learning rate is $8e - 3$, with a decaying factor of 0.6 with patience to be 15 epochs. For MolHIV, all models are trained for 100 epochs, The initial learning rate is $1e - 3$, with a decaying factor of 0.75 with patience to be 15 epochs. The batch size for TUDataset and MolHIV is 32, and 128 for CIFAR10. All student models are trained 3 times and we report the average results with 1 standard deviation. We use Adam optimizer [28] across all the experiments.

**KD Framework.** For GLNN, the strength for $\mathcal{L}_{SL}$ is searched over $\{1.0, 1e - 1, 1e - 2, 1e - 3\}$; For NOSMOG, the strength for $\mathcal{L}_{SL}$ is fixed to 1.0, and the strength for representational similarity loss is searched over $\{1.0, 1e - 1, 1e - 2, 1e - 3\}$, the adversarial feature augmentation is not utilized as in graph classification dataset, node features are typically represented as one-hot vectors with limited dimensions; For MuGSI, the strength for $\mathcal{L}_{SL}$ is fixed to 1.0. $\lambda, \mu, \eta$ in Eq. 11 are searched over $\{1.0, 1e - 1, 1e - 2\}$, $\{1.0, 1e - 1, 1e - 2\}$ and $\{1e - 4, 1e - 5\}$ respectively.

**Other Implementations.** To sample a random walk path for path consistency loss $\mathcal{L}_{\mathcal{P}}$, we use *generate_random_paths* from *networkx*. The path length is fixed to 8 uniformly. The clustering algorithm for $\mathcal{L}_C$ is Louvain method [4]. We use *python-louvain* package for our implementation. The graph pooling function (readout function) is attention-based aggregation [32] or *summation*. Moreover, the graph pooling and cluster pooling share the same pooling function (if attention-based aggregation is utilized) to improve generalizability.

## A.2 Time Complexity Analysis

Although the time complexity during the inference stage for student MLP and GA-MLP models is identical to the vanilla student models without using knowledge distillation. The preprocessing and training stage will incur some extra computational costs. Specifically, in preprocessing stage, to compute $\mathbf{AD}^{-1}\mathbf{X}$ for 1-hop neighborhood aggregation for GA-MLP, it take $O\left(|V|\bar{d}D\right)$ when $\mathbf{A}$ is a sparse

matrix. Here $D$ is the number of feature dimensions and $\bar{d}$ is the average node degree. To compute the clustering assignment, as we use the Louvain method for our implementation, the time complexity is $O\left(|V|log|V|\right)$. The preprocessing is only performed once, hence the cost is affordable. During the training stage, in addition to the training cost of the teacher GNN model and student MLP model, the extra computational cost comes from random walk path sampling for node-level distillation. As *generate_random_paths* from *networkx* follows the implementation of [59], the time complexity is $O(RT \log \bar{d})$, which can be simplified as $O(cRT)$, where $c$ is a small constant, $T$ is the number of steps for each path, $R$ is the number of random walk paths to sample, $\bar{d}$ is the average node degree. Since $T$ and $R$ are typically small integers, the extra computational cost during the training stage is also affordable.

## A.3 Pseudo-code for MuGSI

The pseudo-code of the proposed framework MuGSI is summarized in Algorithm 1.

---

**Algorithm 1** Algorithm for the MuGSI Knowledge Distillation Framework

---

**Input:** Graph datasets $\mathcal{D} = \mathcal{D}_L \cup \mathcal{D}_U$, #epochs $E$, # paths to sample $R$, student model type $\mathcal{M}_S$

**Output:** Predicted labels $\mathcal{Y}_U$ and optimized network parameters of student model $\Theta_S^*$

    Randomly initialize parameters of teacher model $\Theta_T$ and student model $\Theta_S$.

    Train multiple teacher GNN models with different hyperparameters using $\mathcal{D}$, select the best GNN model $\Theta_T^*$.

**for** each graph $G = \{\mathcal{V}, \mathcal{E}, \mathbf{X}\}$ in $\mathcal{D}$ **do**     ▷ Preprocessing stage

    Compute the clustering assignment for each node $v \in G$ using Louvain method

    Compute the top-$k$ non-trivial Laplacian eigenvector $\mathbf{X}_{\text{LaPE}}$

    Set $\mathbf{X} \leftarrow \text{CONCAT}(\mathbf{X}, \mathbf{X}_{\text{LaPE}})$      ▷ Optional

    **if** $\mathcal{M}_S$ is GA-MLP **then**

        Compute 1-hop neighborhood aggregation feature $\tilde{\mathbf{X}} = \mathbf{AD}^{-1}\mathbf{X}$ ▷ For GA-MLP, the student model input is $\mathbf{X}$ and $\tilde{\mathbf{X}}$; for MLP, the model input is $\mathbf{X}$

    **end if**

**end for**

**for** epoch $\in \{1, 2, \ldots, E\}$ **do**

    Obtain $h_G^T$ and $h_G^S$, and calculate whole-graph distillation loss $\mathcal{L}_G$ using Eq. 6

    Obtain $\mathbf{K}_S$ and $\mathbf{K}_T$ according to Eq. 7, and calculate inter-cluster distillation loss $\mathcal{L}_C$ using Eq. 8

    Randomly sample $R$ random walk paths, compute $p(u|v)$ and $q(u|v)$ according to Eq. 9, and calculate path consistency distillation loss $\mathcal{L}_{\mathcal{P}}$ using Eq. 10

    Obtain ground-truth label $\mathcal{Y}_L$ from $\mathcal{D}_L$ and soft logits $\hat{\mathcal{Y}}_L$ from teacher model output, calculate final loss $\mathcal{L}$ using Eq. 11

**end for**

Predict $\mathcal{Y}_U$ from $\mathcal{D}_U$ using optimized student model $\Theta_S^*$

**Return** Predicted labels $\mathcal{Y}_U$ and student model $\Theta_S^*$

---

## A.4 Discussion

In this section, we first establish connections between the subgraph-level distillation loss $\mathcal{L}_C$ and Maximum Mean Discrepancy, then we offer an explanation to explain why MuGSI is effective for graph classification.

*A.4.1 **Relation To Maximum Mean Discrepancy.*** Maximum Mean Discrepancy (MMD) is a widely used criterion in Domain Adaptation [17, 19, 26], which compares distributions in the Reproducing Kernel Hilbert Space (RKHS) [18]. Assume we have two sets of samples, $\mathcal{X} = \{x^i\}_{i=1}^N$ and $\mathcal{Y} = \{y^j\}_{j=1}^M$, drawn from distributions $p$ and $q$, respectively. The squared MMD distance between $p$ and $q$ can be expressed as follows:

$$
\begin{aligned}
\mathcal{L}_{\mathrm{MMD}^2}(\mathcal{X}, \mathcal{Y}) &= \| \frac{1}{N} \sum_{i=1}^N \phi\left(x^i\right) - \frac{1}{M} \sum_{j=1}^M \phi\left(y^j\right) \|_2^2 \\
&= \frac{1}{N^2} \sum_{i=1}^N \sum_{i'=1}^N k\left(x^i, x^{i'}\right) + \frac{1}{M^2} \sum_{j=1}^M \sum_{j'=1}^M k\left(y^j, y^{i'}\right) \\
&\quad - \frac{2}{MN} \sum_{i=1}^N \sum_{j=1}^M k\left(x^i, y^j\right),
\end{aligned}
\tag{12}
$$

where $\phi(\cdot)$ is an explicit mapping function, and $k(\cdot, \cdot)$ is a kernel function that projects the sample vectors into a higher or infinite dimensional feature space. MMD loss is 0 if and only if $p = q$ when the feature space corresponds to a universal RKHS. Minimizing MMD loss is equivalently minimizing the distance between distribution $p$ and $q$. There are many valid kernels for MMD, in the specific case of employing a polynomial kernel $k(x, y) = \left(x^\top y + c\right)^d$ with parameters $d = 2$ and $c = 0$, the resulting MMD is represented as follows:

$$
\mathcal{L}_{\mathrm{MMD}_P^2}\left(\mathbf{H}_C^T, \mathbf{H}_C^S\right) = \|\mathbf{G}_S - \mathbf{G}_T\|_F^2,
\tag{13}
$$

here $\mathbf{H}_C^T \in \mathbb{R}^{N_C \times F}$ and $\mathbf{H}_C^S \in \mathbb{R}^{N_C \times F}$ are the cluster-level representations for a given graph $G$ obtained from teacher and student model respectively, $F$ denotes the hidden dimension size. $\mathbf{G}_S, \mathbf{G}_T \in \mathbb{R}^{N_C \times N_C}$ is the Gram matrix with each entry $g_{ij} = \langle \mathbf{h}_{C_i}, \mathbf{h}_{C_j} \rangle$ (the subscript S and T are omitted here). As illustrated in Eq. 7 and Eq. 8, the inter-cluster distillation loss $\mathcal{L}_C$ is a slightly modified version of $\mathcal{L}_{\mathrm{MMD}_P^2}$, which aims to minimize the distance of the distribution over subgraphs (clusters) for teacher model and student model.

*A.4.2 **Why MuGSI is Effective for Graph Classification.*** The core of MuGSI is the multi-granularity distillation loss, which is composed of graph-level distillation loss, cluster-level distillation loss, and node-level distillation loss. For cluster-level distillation loss $\mathcal{L}_C$, we have shown in the previous subsection that it forces the student model to approximate the teacher GNN model in distributional space over clusters. For graph-level distillation loss $\mathcal{L}_{\mathcal{G}}$, if we assume the graph representations $z_G^T$ and $z_G^S$ for a given graph $G$, as generated by the teacher and student models respectively, follows Gaussian distribution with mean $h_G^T, h_G^S$ and the same covariance $\Sigma$, i.e., $z_G^T \sim \mathcal{N}\left(h_G^T, \Sigma\right)$ and $z_G^S \sim \mathcal{N}\left(h_G^S, \Sigma\right)$, then the KL divergence between $z_G^T$ and $z_G^S$ is given by:

$$
\mathcal{D}_{KL}\left(z_G^T, z_G^S\right) = \frac{1}{2}\left(h_G^T - h_G^S\right)^T \Sigma^{-1}\left(h_G^T - h_G^S\right).
\tag{14}
$$

Consequently, the graph-level distillation loss $\mathcal{L}_{\mathcal{G}}$ serves to minimize this KL divergence, thereby ensuring that the distribution of $z_G^S$ closely approximates that of $z_G^T$. Finally, prior research has established that multi-step random walks are capable of extracting local substructures for any node $v \in G$ [55]. In light of this, we employ random walks to calculate $p(u|v)$ and $q(u|v)$ as a surrogate loss in Eq. 10, thereby aligning the distribution over local substructures between the student and teacher models.

The proposed multi-granularity distillation loss addresses the challenges discussed in Section 1 by generating dense learning signals across multiple scales of graph structure, and ensures a comprehensive transfer of structural knowledge by aligning multiple distributions between the student and teacher models, which is proven to be efficient and effective in extensive experiments.

## A.5 Datasets Statistics

| Dataset | # Tasks | # Graphs | Ave. # Nodes | Ave. # Edges |
|---|---|---|---|---|
| PROTEINS | 2 | 1113 | 39.06 | 72.82 |
| NCI1 | 2 | 4110 | 29.87 | 32.3 |
| BZR | 2 | 405 | 35.75 | 38.36 |
| DD | 2 | 1178 | 284.32 | 715.66 |
| REDDIT-BINARY | 2 | 2000 | 429.63 | 497.75 |
| IMDB-BINARY | 2 | 1000 | 19.77 | 96.53 |
| CIFAR10 | 10 | 60000 | 117.63 | 941.07 |
| MolHIV | 2 | 41127 | 25.5 | 27.5 |

**Table 6: Dataset statistics**

