# OpenReview forum: "MuGSI: Distilling GNNs with Multi-Granularity Structural Information for Graph Classification"
_ACM.org/TheWebConf/2024/Conference — TheWebConf24_

### Official Review · Reviewer_XY1b · 2023-11-16

**Novelty:** 3
**Technical Quality:** 3

**Review:**

This paper studies the problem of knowledge distillation from GNNs to MLPs for graph classification. The major motivation of the proposed method is to enhance the transferred knowledge by introducing the multi-granularity structural information. The paper is overall well-written and easy to follow. The results are encouraging and demonstrate the effectiveness of the proposed method.

Pros:
1. KD from GNNs to MLPs for graph classification is interesting and deserves to be studied.
2. The idea of multi-granularity structural information is interesting.
3. The code is open-sourced.

Cons:
1. The overall technical contribution is limited, although the authors claimed to introduce multi-granularity structural information, they are all represented by naive representation with ReadOut function, which is quite trivial.
2. The way to getting the clusters in the graph is not well described.
3. The node feature augmentation is also trivial and can be treated as a feature engineering.
4. The overall model is a combination of five components, which is hard to optimize.

**Questions:**

1. Is there any specific design of getting multi-granularity structural information rather than naively readout from the node representations?
2. How to get accurate cluster information?
3. How to ensure the stable training and select the best weight parameters?

**Reviewer Confidence:**

4: The reviewer is certain that the evaluation is correct and very familiar with the relevant literature

**Scope:**

4: The work is relevant to the Web and to the track, and is of broad interest to the community

---

### Official Review · Reviewer_Lra7 · 2023-11-21

**Novelty:** 4
**Technical Quality:** 5

**Review:**

This paper proposes a GNN distillation method for graph classification. They use a GNN as the teacher model and take an MLP as the student model, and the optimized loss is composed of three parts: 1) graph-level, 2) subgraph-level, and 3) node-level distillation. Graph-level distillation loss minimizes the distance between the whole graph embeddings from two models; subgraph-level minimizes the inter-cluster loss;  and node-level loss is to keep random-walk path consistency for each node. For the choice of the student model, they take GA-MLP to enhance the expressive power.

Strength:
1. The paper is easy to understand with clear visualization.
2. The study is meaningful since the application of KD frameworks on graph classification is under-explored (as stated by the authors).
3. The three-level distillation strategy is well-aligned with the graph classification problem.

Weakness:
1. The individual components of each distillation loss are not entirely new within the field.
2. The paper presents comparisons with baseline models that also use a GNN-MLP structure. However, the improvements in performance metrics appear marginal across various datasets. It would be better for the authors to include a more in-depth comparison, particularly focusing on inference time, to better understand the practical advantages of the proposed method.
3. Datasets that always be used to test the expressive power should be included, like ZINC.

**Questions:**

Please see the weakness part.

**Reviewer Confidence:**

2: The reviewer is willing to defend the evaluation, but it is likely that the reviewer did not understand parts of the paper

**Scope:**

4: The work is relevant to the Web and to the track, and is of broad interest to the community

---

### Official Review · Reviewer_BM94 · 2023-11-22

**Novelty:** 4
**Technical Quality:** 5

**Review:**

**Summary**

The paper proposes a GNN knowledge distillation framework to improve the expressiveness of MLP on the graph classification tasks. Specifically, the authors design multiple training objectives to align the learned knowledge in different granularities between teacher GNN and student MLP. Finally, the proposed MuGSI framework is evaluated on multiple graph benchmark datasets to demonstrate the effectiveness of distilled student MLP.

**Strengths**

1.	Overall, the paper is well-organized and easy to follow.
2.	Comprehensive experiments are conducted to evaluate the effectiveness of each component.
3.	Enough implementation details are provided.

**Weaknesses**

1.	First of all, I think the motivation of the paper is not clearly clarified. The reason that previous GNN-MLP KD frameworks usually focus on node-level tasks is that the enormous node number in large graphs will significantly increase the memory cost, which is usually not the bottleneck for graph classification task. In this case, what is the motivation of this work if it can not achieve obvious performance improvement than the GNNs? The authors are encouraged to provide further illustrations on this.
2.	Although the proposed framework is evaluated on multiple benchmarks, most of the employed datasets can still not be considered as large datasets. The authors are encouraged to demonstrate the model efficiency on some more large datasets, like ogbg-molpcba and ogbg-ppa.
3.  In this paper, the distribution shift is manually created by removing nodes from the original graph. However, it is not clear how much distribution discrepancy it causes. For graph classification tasks, previous works usually try to create distribution shifts by splitting the dataset according to certain criteria, like molecule scaffold [1, 2]. The authors are encouraged to conduct experiments in this setting.


[1] Graph Contrastive Learning with Augmentations. In NeurIPS 2020

[2] Self-supervised Graph-level Representation Learning with Local and Global Structure. In ICML 2021

**Questions:**

Please refer the weakness section

**Reviewer Confidence:**

4: The reviewer is certain that the evaluation is correct and very familiar with the relevant literature

**Scope:**

4: The work is relevant to the Web and to the track, and is of broad interest to the community

---

### Official Review · Reviewer_wGEG · 2023-11-24

**Novelty:** 4
**Technical Quality:** 5

**Review:**

This paper presents a well-defined problem and proposes a novel solution with extensive experimental validation, demonstrating the effectiveness, efficiency, and robustness of MuGSI. MuGSI introduces a unique approach to knowledge distillation for graph classification, addressing specific challenges and incorporating multi-granularity distillation loss and node feature augmentation. The proposed framework has the potential to advance the field of graph classification by addressing key challenges and improving the performance of student MLPs.

**Questions:**

1.The paper mentions that the superiority of proposed model is demonstrated through experiments on 6 small real-world datasets and 2 large real-world datasets. However, there is no detailed discussion or analysis provided on the characteristics of these datasets or how they were selected.
2.The paper could benefit from a more detailed discussion of the computational complexity associated with the proposed method.
3.It is better to further compare with existing KD frameworks for graph classification.
4. More detailed discussion on the limitations and potential future work could enhance the paper’s impact.

**Reviewer Confidence:**

3: The reviewer is confident but not certain that the evaluation is correct

**Scope:**

3: The work is somewhat relevant to the Web and to the track, and is of narrow interest to a sub-community

---

### Official Review · Reviewer_6ikF · 2023-11-24

**Novelty:** 5
**Technical Quality:** 5

**Review:**

The authors study  a less explored problem: i.e GNN-to-MLP distillation for graph classification. They experimentally study why existing node -classification based methods do not work so well in this setup.
Towards this the authors perform Graph level, subgraph level and node -level distilation to tackle this problem.

The idea is interesting, simple to implement and detailed description has been provided.

The authors have conducted a significant number of experiments on a variety of datasets.
The gains look good on a variety of datasets and teacher GNN architectures.

The authors also conduct additional study when graph is modified.

Further, the inference speed of proposed model(onCPU) is faster than existing methods( on CPU, GPU).

Code is also shared.

Overall, a good work!

**Questions:**

1.
The authors mention this:

" Sparse learning signals. For node classification, dense
learning signals can be generated through node-level gradient updates
using soft labels, especially for large-scale graphs that consist
of thousands or even millions of nodes. Conversely, graph classification
inherently provides sparse learning signals, as soft labels are
obtained at the level of entire graphs, making the KD process for
graph classification more challenging"

Is there any citation/study to support this?


2 ".However, this condition is often not met in graph classification
tasks due to the limited input feature space, which severely limits
the expressive power and learning capability of student MLPs. "

Needs more clarification. What is limited input feature space in this context?

3. Which pooling function is used in which result instance? Can the authors mention it in the revision? How is the pooling function chosen?
How does the performance vary(for diff methods and datasets) with diff pooling functions.

**Reviewer Confidence:**

3: The reviewer is confident but not certain that the evaluation is correct

**Scope:**

4: The work is relevant to the Web and to the track, and is of broad interest to the community

---

### Decision · Program_Chairs · 2024-01-22

**Decision:**

Accept

**Comment:**

This paper presents an approach to knowledge distillation from GNNs to MLPs for tasks beyond node classification. While one of the reviewers disagreed with the others on the novelty of the work, overall this work is a good technical contribution and pushes the state of the art in knowledge distillation of GNNs.